



# A comprehensive assessment of in situ and remote sensing soil moisture data assimilation in the APSIM model for improving agricultural forecasting across the U.S. Midwest

Marissa Kivi[1], Noemi Vergopolan[2], Hamze Dokoohaki[1*]

1 Crop science department, University of Illinois at Urbana-Champaign, Urbana, IL, USA
2 Department of Civil and Environmental Engineering, Princeton University, Princeton, NJ, USA

*Corrpospoding author: Hamze Dokoohaki; hamzed@illinois.edu

**Abstract.** Today, the most popular approaches in agricultural forecasting leverage process-based crop models, crop monitoring data, and/or remote sensing imagery. Individually, each of these tools has its own unique advantages but is, nonetheless, limited in prediction accuracy, precision, or both. In this study we integrate in situ and remote sensing (RS) soil moisture observations with APSIM model through sequential data assimilation to evaluate the improvement in model predictions of downstream state variables across 5 experimental sites in the U.S Midwest. Four RS data products and in-situ observations spanning 19 site-years were used through two data assimilation approaches namely Ensemble Kalman Filter (EnKF) and Generalized Ensemble Filter (GEF) to constrain model states at observed time steps and estimate joint background and observation error matrices. Then, the assimilation's impact on estimates of soil moisture, yield, NDVI, tile drainage, and nitrate leaching was assessed across all site-years. When assimilating in situ observations, the accuracy of soil moisture forecasts in the assimilation layers was improved by reducing RMSE by an average of 17% for 10cm and ~28% for 20 cm depth soil layer across all site-years. These changes also led to improved simulation of soil moisture in deeper soil layers by an average of 12%. Although crop yield was improved by an average of 23%, the greatest improvement in yield accuracy was demonstrated in site-years with higher water stress, where assimilation served to increase available soil water for crop uptake. Alternatively, estimates of annual tile drainage and nitrate leaching were not well constrained across the study sites. Trends in drainage constraint suggest the importance of including additional data constraint such as evapotranspiration. The assimilation of RS soil moisture showed weaker constraint of downstream model state variables when compared to the assimilation of in situ soil moisture. The median reduction in soil moisture RMSE for observed soil layers was lower, on average, by a factor of 5. However, crop yield estimates were still improved overall with a median RMSE reduction of 17.2%. Crop yield prediction was improved when assimilating both in-situ and remote sensing soil moisture observations and there is strong evidence that yield improvement was higher when under water-stressed conditions. Comparisons of system performance across different combinations of remote sensing data products indicated the importance of high temporal resolution and accurate observation uncertainty estimates when assimilating surface soil moisture observations.

**Keywords**: Model-data integration, Sequential Data Assimilation, APSIM, soil moisture



## 1. Introduction

To effectively address pressing global food security challenges, agricultural forecasting tools must exhibit high
accuracy and precision across spatial and temporal scales. As process-based crop models offer a system-level
representation of many soil and crop processes, they are increasingly recognized as practical forecasting tools in
agricultural research (Silva and Giller, 2021; Fer at al., 2021). However, their weakness comes from many
unaccounted uncertainties, such as those related to model parameters, initial conditions, and weather (Dokoohaki et
al., 2021). Prior studies have shown state data assimilation (SDA) to be a powerful tool to overcome this weakness in
process-based crop models (e.g. Kivi et al., 2022, Dokoohaki et al., 2022a). SDA enables a temporally-continuous,
high-dimensional scaffold in which a variety of observations can be smoothly integrated using one of many robust,
systematic algorithms, such as the Ensemble Kalman Filter (EnKF; Dietze et al., 2017; Huang et al., 2019; Liu et al.,
2021; Dokoohaki et al., 2022a; Kivi et al., 2022). Through SDA, uncertainty around spatially-heterogenous and
dynamic properties in agricultural systems can be constrained, thereby increasing precision and accuracy in estimates
while decreasing dependence on extensive site-level model calibration (Mishra et al., 2021).
Numerous past studies have used SDA to constrain crop model estimates, using observations on leaf area index (e.g.,
Nearing et al., 2012; Ines et al., 2013; Ma et al., 2013; Chen et al., 2018; Lu et al., 2021), soil moisture (Kivi et al.,
2022), biomass (e.g., Linker and Ioslovich, 2017) and evapotranspiration (e.g., Huang et al., 2015). For example, a
synthetic study by Zhu et al. (2017) found that the assimilation of coarse resolution surface soil moisture data into a
coupled soil water-groundwater numerical model constrained soil moisture estimates in the first 50 cm of the soil
profile despite explicitly unaccounted spatial heterogeneity in soil properties. These studies showed how SDA can
partially account for the spatial variability in soil hydraulic conductivity across broad regions without explicit model
calibration. In addition to incorporating spatial heterogeneity in soil properties, Kivi et al. (2022) demonstrated that
the assimilation of high quality and frequent in-situ soil moisture observations can substantially improve downstream
model predictions of tile drainage, nitrate (NO3) leaching, and root-zone soil moisture (RZSM) for maize and
soybeans in the APSIM model. However, collecting field measurements of soil moisture for different cropping
systems, soils, and environments is expensive, extremely laborious, and time-consuming.
Alternatively, the assimilation of high-resolution Remote Sensing (RS) data products dramatically increases SDA
applications' range beyond in situ data availability by effectively capturing the spatiotemporal variability of many
agricultural state variables, such as vegetation cover and soil moisture, with consistency and high temporal frequency
(Peng et al., 2017). As a result, RS observations could be invaluable to constraining model predictions at the regional
scale and have been increasingly applied for agricultural forecasting in the data assimilation literature, as demonstrated
in literature reviews by Dorigo et al. (2007), Huang et al. (2019), and Weiss et al. (2020). The application of RS soil
moisture data products has been especially popular and successful in data assimilation-focused agricultural forecasting
studies. These data products, which characterize soil moisture content in the first 5 cm of the soil profile, pull
information from active and/or passive sensors of microwave reflectance. Due its high sensitivity to surface soil
moisture, many data products have been developed around available L-band microwave sensor information collected
by NASA's SMAP Mission (Kumar et al., 2018). The SMAP-HydroBlocks data products merges SMAP data with
the HydroBlocks land surface model to increase spatial resolution in the final estimates and improve scalability



(Vergopolan et al., 2021b), while the SMAP-Sentinel1 data product pairs SMAP data with Sentinel-1 radar
information to achieve similar goals (Das et al., 2019). Others, like the ESA-CCI data product (Dorigo et al., 2017),
compile information from multiple sensors, including the SMAP passive sensor, to allow for greater temporal
coverage. However, this approach comes at the cost of coarser spatial resolution.
Nonetheless, as demonstrated in past studies, the assimilation of RS soil moisture data has its limitations. First,
uncertainty and biases in RS data products are typically poorly defined (Huang et al., 2019). RS-based data products
are based on empirical relationships, and, as they are predicted as a function of surface reflectance, uncertainties in
the raw radiance will propagate unsupervised into final estimates (Weiss et al., 2020). Additionally, RS estimates
characterize soil moisture in only the top 5 cm of the soil profile and, thus, rely on models or empirical
parameterizations to describe the root zone soil profile. Among others, De Lannoy et al. (2007) and Monsivais-
Huertero et al. (2010) both found the assimilation of in-situ near-surface soil moisture observations to be far less
effective than that of in-situ root-zone soil moisture observations in constraining estimates of the greater soil water
profile. Yet, since the surface layer is typically the layer where fertilizers are added, the accurate estimation of surface
layer state variables is essential for today's agroecosystems (Verburg and CSIRO, 1996). To overcome relatively
coarse spatial resolution in RS data products, past studies have explored downscaling approaches (e.g., Chakrabati et
al., 2014) or leveraged additional in-situ datasets (e.g., Liu et al., 2021) to overcome "mismatch" challenges and
downscale RS soil moisture estimates to more accurately reflect field scale measurements (Vergopolan et al., 2021a).
However, the reliance on in situ observations of these approaches can limit system transferability across broad regions
(Peng et al., 2017). Moreover, as described by Crow et al. (2012), it can be difficult to properly evaluate coarse soil
moisture estimates with point-scale ground measurements due to unknown and often significant sampling uncertainty.
Data assimilation with process-based models has been previously applied as a robust and scalable way to leverage
information in coarse resolution soil moisture estimates (e.g. Vergopolan et al., 2021b).
Despite the immense theoretical potential of SDA with both in situ and RS observations, past studies have reported
inconsistent SDA performance in modeling crop yields. For example, de Wit and van Diepen (2007) observed
inconsistencies in yield constraint when assimilating soil wetness index (SWI) derived from 0.25° ERS1/2 microwave
radiance information into the WOFOST model across agricultural regions of Spain, Germany, France, and Italy. They
partially attributed poor predictions in certain regions to irrigation processes that were not captured by the model nor
coarse resolution SWI observations. Lu et al. (2021) also saw year-to-year variability in assimilation performance
when assimilating in situ observations of canopy cover and soil moisture for 6 site-years in Nebraska. When
assimilating soil moisture independently, canopy cover estimates were better constrained in drier years. They
suspected this to result from the canopy cover's lower sensitivity to soil moisture in the model when water is in surplus
(i.e., due to energy-limited conditions). We further suspect that SDA's inconsistent performance is related to the
misrepresentation of model processes linking soil moisture to crop- and soil-related variables (e.g., soil nitrogen, leaf
expansion, crop water uptake). As a result, direct upstream improvement of model state variables with SDA does not
always translate into improvement in downstream results. To understand the role of soil moisture data assimilation in
improving crop yields and better pinpoint areas for future improvement, a comprehensive assessment that investigates
performance across time and different genetic (G), environmental (E), and management (M) spaces is required.



Although a growing body of studies has attempted to quantify the impact of soil moisture assimilation in crop models, such a comprehensive evaluation of in situ and RS soil moisture SDA in crop models across GxExM spaces is still lacking (Folberth et al. 2016b; Kivi et al., 2022).

To bridge this knowledge gap, we present a comprehensive assessment of soil moisture data assimilation as a method for constraining crop model predictions across the U.S. Midwest. Building on the assimilation framework in Kivi et al. (2022), we independently assimilated both in situ and RS soil moisture observations in the APSIM crop model at five experimental sites in the U.S Midwest. With field data covering 19 site-years of corn and soybean cropping systems across the region, this study tests the data assimilation system across a broader GxExM inference space and quantifies the benefit of assimilating different RS soil moisture products in comparison to the in-situ soil moisture observations. The main objectives of this study were:

1. To quantify how in situ soil moisture observations can constrain crop model forecasts of downstream estimates, including root-zone soil moisture, crop yield, crop phenology via NDVI, tile drainage flow, and NO3 leaching through SDA.
2. To quantify the added benefit of RS soil moisture observations in improving crop model predictions of root-zone soil moisture, crop yield, and crop phenology via NDVI through SDA.

## 2. Methods

Sections 2.1 and 2.2 describe the five experimental sites and the in-situ observations employed in this study for model set-up, SDA, and evaluation. Section 2.3 outlines the four different RS soil moisture data products that were assimilated, and Section 2.4 presents the data-assimilation system introduced in Kivi et al. (2022). Sections 2.4.1-2.4.4 highlight the improvement made to the system presented by Kivi et al. (2022) that were applied in this work, and Section 2.4.5 defines the different simulation experiments performed.

### 2.1 Study sites

This study focused on five experimental sites across the U.S. Midwest with in-situ observations of soil moisture, crop yield, nitrate load, and tile drainage flow for 19 years between 2011 and 2019. Site IL was the Energy Farm, a well-monitored experimental site in central Illinois that was the focus of the development and initial evaluation of the employed data-assimilation system (Kivi et al., 2022). Site IN, MN, OH, and SD were available through the Transforming Drainage (TD) project (Chighladze et al., 2021). The TD project database is publicly-available and contains high-quality data from 39 tile-drained research sites with data spanning over 200+ site-years. The available observations include data on tile drainage, yield, water table, water quality, and soil characteristics, among many others. Though numerous sites were available as part of the project, the experimental design and data available for each site-year varies widely in the database. For consistency, this work required that each site-year include a plot with: (1) a free tile drainage system, (2) available NO3 load and tile flow data at the plot level, (3) available in situ soil moisture observations, (4) maize or soybean crops, and (5) a rain-fed system. We identified only 17 site-years across five sites in the database which satisfied all these criteria.





To properly set up the APSIM model for each of the five sites, we included all available site information on each year,
cropping system, residue type, planting and harvesting details, tillage practices, and fertilizer applications as constants
in the simulations. Following updated information available through Moore et al. (2021), the IL setup of Kivi et al.
(2022) now includes tillage practices in the model set-up and increased nitrogen (N) fertilizer from 64.6 kg N/ha, to
202 kg N/ha. Detailed information on the plot and management information for all five sites are included in the
Supplementary Materials (Table A1). Study sites will be referred to by their given study IDs in Figure 1.
**2.2 Observation data**
*In situ soil moisture*
Across the study site-years, sub-daily soil moisture (SM) observations were collected at various soil depths between
10 and 105 cm using soil sensors; the measured depths and sensor type varied by site. All observations are available
in units of volumetric water fraction (VFW; mm/mm). For the 4 TD sites, SM observations were only available as
daily averages. For consistency, SM observations at IL (available at 15-minute intervals) were aggregated to daily
averages when at least 40 15-minute observations were available. Observations from the winter months (December-
March) were excluded due to the influence of freezing soils. Across all site-years, in situ SM assimilation was
performed with available observations for the 10- and 20-cm soil depths, which hereinafter will be referred to as SM3
and SM4, respectively. All other available SM observations for deeper soil layers were used to evaluate model root-
zone SM estimates. SM observations were paired with an APSIM soil layer based on the recorded sensor depth and
the site soil profile. In the case that more than one observation was available for a given APSIM soil layer, the average
SM was computed for each day and layer with the assumption of uniform SM in the layer.

*Harvested maize and soybean yields*
Data on harvested yield for the TD sites were available for each site-year with 1-3 replicated measurements. These
replicated observations were averaged and converted from grain at standard moisture content (i.e., 15.5% for maize
and 13% for soybean) to dry-grain weight for best comparison with the APSIM model output. Observations for IL
were already recorded as dry-grain weights and given in units of kg/ha (Kivi et al., 2022). Across 12 maize site-years,
observed yields ranged from 6.51 to 13 Mg/ha with an average yield of 9.93 Mg/ha. The 7 soybean site-years had
observed yields ranging from 2.78 to 4.15 Mg/ha with an average yield of 3.50 Mg/ha.

*Remotely sensed Normalized Difference Vegetation Index (NDVI)*
The normalized difference vegetation index (NDVI) can be used to quantify vegetation greenness and reasonably track
the phenological development of crops (Gao and Zhang, 2021). In this study, NDVI observations from Landsat
between 2011 and 2019 were used to evaluate APSIM's performance in predicting crop phenology for each site-year.
NDVI time series were extracted at each site location from Landsat 7 and 8 remote sensing imagery courtesy of the
U.S. Geological Survey via Google Earth Engine and derived from the red (RED) and near-infrared (NIR) spectral
bands using the following equation:


$$NDVI = \frac{NIR - RED}{NIR + RED} \tag{1}$$


*In situ measurements of tile drainage and nitrate load*

Daily observations of tile drainage flow (mm) and NO3 load (kg NO3-N ha-1) were available for all 19 site-years.
Any missing daily drainage values for the TD sites had been imputed previously and used to approximate missing
values of daily NO3 load, as described by Helmers et al. (2022). Methods and instrumentation used to collect and
process the TD sites and IL data are presented by Helmers et al. (2022) and Kivi et al. (2022), respectively. In this
study, daily values for tile drainage flow and NO3 load were summed to annual values for comparison with model
output. For the purposes of this analysis, we assumed any day with NA tile drainage flow values in the data had no
drainage and no NO3 loss.

**2.3 Remote sensing soil moisture**

To assess the performance of SM data assimilation with satellite-based observations, we included 4 RS data products
that span different temporal and spatial resolutions (Table 1). These observations were extracted at the point level for
the study sites and serve to represent the first 5 cm of the soil profile or surface SM. Observations from the winter
months (i.e., December-March) were removed to avoid issues with snow cover and freezing soils. The product IDs
provided in Table 1 will be used to identify each data product.

*ESA-CCI*

The RS dataset with the coarsest spatial resolution in this study was the ESA-CCI SM product. Each year, the European
Space Agency Climate Change Initiative (ESA CCI) algorithmically merges information from 3 active (e.g., ASCAT
A/B) and 10 passive (e.g., SSM/I, AMSR-E, SMOS, SMAP) microwave sensors to estimate daily surface SM globally
for over 40 years. Dorigo et al. (2017) provide complete documentation on how these data products are produced.
Here we used the combined product (version v06.1), which includes daily uncertainty estimates. Several past studies
have assimilated this data product into process-based models with varying levels of success (e.g., Zhou et al., 2016;
Liu et al., 2017; Liu et al., 2018; Naz et al. 2019).

*SMAP-HydroBlocks*

The SMAP-HydroBlocks surface SM dataset has the highest spatial resolution in this study. It was introduced by
Vergopolan et al. (2021b) by combining the HydroBlocks land surface model, a Tau-Omega radiative transfer model,
machine learning, in situ SM observations, and SMAP remotely sensed satellite observations to estimate surface SM
with 30-meter resolution across the contiguous United States. In specific, the Hydroblocks model was coupled with a
Tau-Omega radiative transfer model (HydroBlocks-RTM) and used to simulate SM, soil temperature, and brightness
temperature at a 3-hour, 30-meter resolution. Brightness temperature estimates from NASA's Soil Moisture Active
Passive (SMAP) mission were then merged with the HydroBlocks-RTM estimates using a spatial cluster-based
Bayesian merging scheme (Vergopolan et al., 2020). Using the inverse HydroBlocks-RTM, SM was estimated at





SMAP overpass time at 30-m spatial resolution. Vergopolan et al. (2021b) reported an RMSE of 0.07 mm3/mm3 after
comparing SMAP-Hydroblocks estimates to in situ observations from 233 independent experimental sites. This study
is the first to assimilate SMAP-HydroBlocks SM estimates into a crop model. SM morning and afternoon retrievals
were aggregated to a daily resolution, and site-level estimates were computed as the mean value of any data point
within 0.0005° of the given site location. The uncertainty estimate for each observation was calculated based on the
spatial variability of selected data points for that time step and the reported standard error (SE = 0.07 mm3/mm3) as :

$$Var(Y_{s,t}) = Var(y_t) + SE^2 \qquad (2)$$


where, for site s at the tth available time step, Y represents the site-level SM estimate, and y presents SM estimates
within 0.0005° of the site location.

*SMAP-Sentinel1*
The SMAP-Sentinel1 SM product was produced by merging information collected by the SMAP L-band radiometer
and the Copernicus Project Sentinel-1 C-band radar. After the malfunction of the SMAP radar in 2015, Sentinel-1
active microwave data were used with passive microwave sensor information from the still-operating SMAP
radiometer to estimate surface SM content globally using the active-passive algorithm. Although the merged product
increased the revisit interval from 3 to 12 days, it enabled retrievals at two different spatial resolutions (i.e., 1 km and
3 km; Lievens et al., 2017). Upon comparing the estimates with in situ SM measurements, Das et al. (2019) reported
RMSE for SMAP-Sentinel1 SM estimates as roughly 0.05 m3/m3. In this study, this value was applied as the standard
error for SM estimates at both spatial resolutions and at all available time steps. Retrievals were available for all TD
site-years but were unavailable for IL for unknown reasons.
**2.4 Data-assimilation system**
This study uses the data-assimilation system developed and evaluated in Kivi et al. (2022). The original system
leveraged the pSIMS platform, APSIM crop model, Ensemble Kalman Filter (EnKF), and an algorithm presented by
Miyoshi et al. (2013) to estimate and propagate uncertainties, perform sequential data assimilation, and generate daily
agricultural forecasts at the field scale. The following sections provide details on the new development and advances
in the Kivi et al. (2022) approach. The workflow is illustrated in Figure 2. APSIM management variables that were
known include planting and harvest dates, fertilizer amount, type, and timing, tillage type, depth, and timing, crop
type, row spacing, sowing density, and, if available, planting depth.
**2.4.1 Model parameter priors**
Initial soil water, cultivar, and residue weight were randomized across model ensembles for each site to incorporate
uncertainty around initial conditions. If unavailable in the management data, planting depth was also randomized and
drawn from different prior distributions for each crop. These distributions represented reasonable planting depth
ranges for the two crops in the Midwest, as described in extension websites produced by the University of Missouri



(Luce, 2016) and Michigan State University (Staton, 2012). Using a uniform prior distribution, planting depths ranged
from 1.5 to 2.5 inches for maize and 1 to 2 inches for soybean.
Prior distributions were also set to incorporate uncertainty around cultivar. For maize, nine cultivar parameters were
ensembled, including the six cultivar parameters (i.e., tt_flower_to_maturity, tt_flower_to _start_grain,
tt_maturity_to_ripe, tt_emerg_to_endjuv, head_grain_no_max, grain_gth_rate) randomized in Kivi et al. (2022). The
other three parameters (i.e., largestLeafParams1, leaf_init_rate, leaf_app_rate1) were drawn from Dokoohaki et al.
(2022b), who identified maize cultivar parameters that were influential for estimates of leaf area index (LAI) in the
APSIM Maize module and optimized their value distributions using a hierarchical Bayesian optimization approach
across the U.S. Midwest. Table A.2 gives more detailed information on all randomized parameters and their prior
distributions. We completed a preliminary assessment of the Maize module at each of the study sites and found that,
under the given parameter value ranges, APSIM was capable of appropriately simulating the phenological
development and grain yield for maize at each site.
The selection of soybean cultivars for each site was determined using a semi-systematic approach. First, a range of
maturity groups was determined for each site based on a study by Mourtzinis and Conley (2017), which delineated
soybean maturity groups across the U.S. We defined the upper and lower maturity group bounds for each site using
the bounding zone contour lines for each site location in Figure 4 of Mourtzinis and Conley (2017). Then, initial
APSIM simulations were performed for each site using all APSIM-defined soybean cultivars falling within the
prescribed maturity group range. The model results were compared to the observed soybean yields at each site, and
the best-performing maturity group (MG) for each site was determined. The final range for each site was
approximately MG ± 0.5. In each ensemble, the cultivar for each crop at each site was assumed to be constant across
all site-years.
**2.4.2 Weather and soil model drivers**
To incorporate uncertainty around soil and weather into our simulations, a Monte Carlo sampling approach was used
to randomly assign ensembles of weather and soil drivers to model ensembles. For each study site, ten weather
ensembles from the ERA5 reanalysis data product were employed to characterize solar radiation, maximum air
temperature, minimum air temperature, precipitation, and wind speed at the daily resolution and at each site location.
ERA5 is a global gridded reanalysis data product from the European Centre for Medium-Range Weather Forecasts
(ECMWF), which characterizes the weather state variables at hourly time steps with associated uncertainties
(Hersbach et al., 2020). In addition, 25 soil ensembles were generated from the SoilGrids global gridded soil database
(Hengl et al., 2014) for each site location. These ensembles cover 30 soil properties (including available water lower
limit, bulk density, drained upper limit, organic carbon, soil class, and pH) and were created by sampling from each
soil parameter mean and uncertainty values available in the SoilGrids dataset.
**2.4.3 PROSAIL model**
Since APSIM does not currently estimate NDVI, APSIM was coupled with the PROSAIL model described in
Dokoohaki et al. (2022b) to estimate daily NDVI values and enable the appropriate evaluation of the model's
simulation of crop phenology at the study sites. The PROSAIL model is a radiative transfer tool that combines
PROSPECT, a leaf optical properties model, and SAIL, a canopy bidirectional reflectance model, to estimate spectral
reflectance for a given vegetative area based on soil and plant/canopy properties (Jacquemoud et al., 2009). In this
study, APSIM's daily forecasts of soil and plant variables were transformed and used as inputs into the PROSAIL
model to compute the spectral reflectance for each ensemble. Then, for each day and ensemble, the estimated spectral
information was used to estimate NDVI using the vegetation index function within the hsdar R library (Lehnert et al.,
2019). Further details on the coupling protocols can be found in Dokoohaki et al., (2022b).

**2.4.4 Ensemble Kalman filter with the Miyoshi algorithm**
The data-assimilation system presented in Kivi et al. (2022) (which we will call EnKF-Miyoshi hereinafter) employs
the ensemble Kalman filter (EnKF) to assimilate SM observations into the APSIM model. The EnKF merges
information from the model ensemble forecast distribution and observations (with associated uncertainty) at each time
step to optimally estimate the state of the system (Evensen, 2003). The system also leverages the Miyoshi algorithm
in series with the EnKF to improve estimates of the two system uncertainty matrices (i.e., Pf and R) and improve filter
performance. Based on diagnostic innovation statistics, the Miyoshi algorithm estimates a forecast inflation scalar (Δ)
and observation uncertainty (R) at each analysis time step. At time step t with available data, the system follows the
following steps:
1. The mean (Xf,t) and the variance-covariance matrix (Pf,t) of the model forecast ensemble are computed to
define the forecast distribution, which is assumed to follow a Normal distribution.

2. The observed distribution (Yt) is also assumed to be Normal with mean yt and variance-covariance matrix
Rt, where Rt = R* from the previous analysis time step or R1 = Σ. Σ is a diagonal matrix that assumes 10%
standard error for each observed state variable.

3. The Kalman Gain (K) is computed as follows, where Δt = Δ* or Δ1 = I (I is the identity matrix) and H is the
observation operator:

$$K_t = \Delta_t P_{f,t} H^T (R_t + H \Delta_t P_{f,t} H^T)^{-1} \tag{3}$$


4. The analysis distribution, which assumes a Normal distribution, is determined with mean (Xa,t) and
variance-covariance matrix (Pa,t).

$$X_{a,t} = X_{f,t} + K_t (Y_t - H X_{f,t}) \tag{4}$$
$$P_{a,t} = (I - K_t H) P_{f,t}$$


5. The model ensemble is updated at each time step according to the analysis distribution based on each
ensemble's likelihood within the forecast distribution.

6. Δ* and R* are recomputed using the following series of equations, where do-a and do-f represent the
observation-analysis and observation-forecast innovations for the current time step, respectively, E denotes
the expectation operator, and ρ is a user-defined weight given to the new estimate. A lower bound of 1 is
imposed on each entry in Δest and only the diagonal entries of Rest are maintained.





$$\mathrm{E}\left(d_{o-a}d_{o-f}^{T}\right) = R_{est}$$

$$\Delta_{est} = \frac{d_{o-f}^{T}\,d_{o-f}\ -\ R_{est}}{H\Delta_{t}P_{f,t}H^{T}}$$

$$R^{*}\ =\ (\rho)R_{est}\ +\ (1-\rho)R_{t}$$

$$\Delta^{*} = (\rho)\Delta_{est}\ +\ (1-\rho)\Delta_{t} \tag{5}$$


### 2.4.5 Generalized ensemble filter


However, the EnKF-Miyoshi workflow as established cannot robustly handle observation operators (H) that change
dimensions over time. However, to reduce information loss within the system, H must be able to adapt according to
the number of observations available. To increase flexibility in system configuration, an alternative sequential data
assimilation approach was tested in this work to replace the EnKF-Miyoshi method. The new method, hereinafter
called the Generalized Ensemble Filter (GEF), comprises a fully numerical Bayesian approach to estimating the
analysis distribution and an inflation scalar. The model resembles the approach presented by Raiho et al. (2020) and
Dokoohaki et al., (2022a) and has the following form at analysis time step t:

$$Q \sim U(0.001, 5)$$

$$X_{A} \sim N(X_{f,t},\ P_{f,t}\ +\ (Q-1)*diag(P_{f,t})) \tag{6}$$

$$Y_{t} \sim N(X_{A}, R_{t})$$


where Q is the estimated forecast inflation scalar and XA is a drawn sample from the analysis distribution. The
estimation of XA and Q was completed using a Markov Chain Monte Carlo (MCMC) approach by leveraging the
nimble R library (de Valpine et al., 2017). Though not explored in this study, this approach also allows for the
definition and estimation of more complex relationships between observations and model forecasts (e.g., nonlinear
observation operators).
In this study, the GEF was applied over the EnKF-Miyoshi workflow when (1) more than one observation was
assimilated for a single state variable at a given time step or (2) the number of available observations varied throughout
a simulation (i.e., changing H). Conversely, the GEF approach was ineffective for cases where only one observation
was available at a given time step, as the MCMC algorithm did not converge due to limited data. The EnKF-Miyoshi
was applied in these settings.

### 2.4.6 Simulation schemes


All simulations in this study were performed with 100 ensembles and with a 4-month initialization period starting on
1 Jan of the first year at each site. There were nine different simulations performed for each site in this study which
varied in terms of observations assimilated and assimilation method applied. First, two "baseline" runs were completed
across all 19 site-years to establish system performance benchmarks. As a lower bound on performance, a free model
simulation was performed with no data assimilation. To set an upper bound, SM sensor observations were assimilated
into the model to represent an "ideal" SM data assimilation setting. Next, two groups of runs were performed to test
the assimilation of RS SM data products: "individual" and "additive" runs. In the "individual" runs, all 4 RS data





products were assimilated independently within the system. These runs were performed to compare the value of
different RS data products directly. Then, in the "additive" runs, observations from multiple RS data products were
jointly assimilated into the system following an additive approach. The first iteration included only ESA observations,
and each subsequent iteration added another data product until all 4 data products were included (i.e., ALL). Data
products were added in succession based on availability, such that the first data product tested had the highest average
number of observations per year. By sequentially adding new data products, the additional impact of each RS data
product could be evaluated. To allow for the application of the GEF in runs with more than one data product, a
minimum of 2 observations per day were required for the "additive runs" to ensure the convergence of the MCMC
algorithm. For all runs where RS data were assimilated, only site-years after 2014 were investigated due to the limited
temporal extent of RS data products.
**2.5 System evaluation**
This study applied the year-average ensemble weighting strategy, as presented in Kivi et al. (2022), to leverage all
available information from the simulations and evaluate the results more accurately. In each site-year simulation, daily
weights were assigned to each ensemble as the likelihood of producing the daily estimate given the analysis
distribution, and ensemble weights were normalized across the model ensemble for each day. Finally, the average
annual weight for each ensemble was computed for each site-year. The application of annual weights in the analysis
was the most robust for evaluating yearly estimates (e.g., yield, cumulative NO3 load, cumulative tile drainage).
To evaluate the accuracy and precision of model forecasts for each site-year simulation, we utilized the root mean
squared error (RMSE), spectral norm, and weighted variance. RMSE was calculated for each run to quantify changes
in accuracy between runs, while the spectral norm and weighted variance were employed to quantify changes in
precision (Kivi et al., 2022). Additionally, to help standardize accuracy measures across site-years, a normalized
RMSE (nRMSE) was calculated as :

$$nRMSE\ (\%) = 100 * \frac{RMSE}{\overline{Y}} \tag{7}$$

where $\overline{Y}$ is the average observed value. Changes in accuracy and precision between the free model and SDA were
quantified by computing the relative change in each metric for the two runs. For example, for calculating the change
in RMSE, we computed :

$$\Delta RMSE\ = \frac{RMSE_{SDA} - RMSE_{FREE}}{RMSE_{FREE}} \tag{8}$$

The coefficient of determination (R2) was used to compare model performance for each state variable more effectively
across all observed time points. It was calculated as :

$$R^2 = 1 - \frac{\sum_{t=1}^{T}(Y_t - \bar{X}_t)^2}{\sum_{t=1}^{T}(Y_t - \bar{X}_t)^2 + \sum_{t=1}^{T}(\bar{X}_t - \overline{Y})^2} \tag{9}$$

where Yt is the observed value at the tth observed time step and is the simulated weighted mean at the tth observed
time step. All observations (n = T) from all site-years were included in this calculation. Separate R2 values were
computed for the Free and SDA results. Weighted mean estimates were computed using annual ensemble weights.





To identify and quantify relationships between variables, one of two correlation statistics was employed
depending on the sample size of the data. When comparing data with a sufficiently large sample size (n > 30), the
Pearson correlation coefficient (r) was calculated to determine the direction and strength of the linear relationship
between two variables.

$$r = \frac{\sum_{i=1}^{n}(x_i - \bar{x})\,(y_i - \bar{y})}{\sqrt{\sum_{i=1}^{n}(x_i - \bar{x})^2} * \sqrt{\sum_{i=1}^{n}(y_i - \bar{y})^2}} \qquad (10)$$

When comparing data at the site-level (n ≤ 19), the Spearman rank-order correlation coefficient (rs) was applied,
which is a nonparametric measure of the strength and direction of the monotonic relationship between two variables.
Though the sample size in this case is still too small for proper application, the Spearman coefficient was applied as
its assumptions are less strict than the Pearson coefficient. It is calculated as :

$$r_s = 1 - \frac{6\sum_{i=1}^{n} d_i^2}{n\,(n^2 - 1)} \qquad (11)$$

where the di is the distance between the two ranks of the ith complete pair (i.e., xi and yi). For both coefficients, a test
for association between paired samples was used to determine significance.

**3. Results**

The results in section 3.1 evaluate the forecast accuracy and precision of in situ SM SDA in comparison to
the free model. Section 3.2 investigates changes in forecast accuracy and precision when assimilating SM RS
observations. The individual runs are assessed with regard to their data characteristics (i.e., retrieval interval and single
vs. multi-sensor development), and the additive runs are evaluated in succession to determine the relative impact of
added observations. Lastly, the impact of RS-based SDA on the forecast accuracy and precision of state variables is
investigated and compared.

**3.1  Assimilation of in situ soil moisture**

**3.1.1 Impact on soil moisture**

Across all assimilation time steps, the free model tended to overpredict SM within the two assimilation layers
(Fig. 3). Therefore, the adjustment in the SDA analysis step typically reduced the total amount of water in the soil
profile. In SM forecasts for the two assimilation layers (i.e., SM3 and SM4), SDA performed as well or better than
the free model in accuracy across all site-years. The median change in RMSE due to SDA was -17% and -28% for
SM3 and SM4, respectively (Fig. 4). Average forecast precision for SM3 and SM4 was also increased with SDA in
84% of cases and by 23% on average.
The three site-years where precision was not increased in SDA include OH in 2013 and 2014 and MN in
2013. Interestingly, these site-years were among those with the most remarkable improvement in accuracy. This
relationship is intuitive considering the nature of the Miyoshi algorithm, which systematically inflates model forecast
uncertainty at time steps when observed and forecasted SM distributions differ substantially. At the cost of reduced





forecast precision, such inflation allows for the filter to pull the model forecast toward the observed distribution and
improve accuracy in future predictions.
SDA's constraint of SM3 and SM4 also led to the indirect constraint of SM in deeper soil profile layers. Across all
site-years with available data, the median change in RMSE for SDA estimates of SM5, SM6, and SM7 was -14%, -
8%, and -14%, respectively. For each of these state variables, SDA increased RMSE for 1-2 site-years, but most site-
years showed improvement or similar performance when compared to the free run. In terms of precision, SDA had an
overall positive impact on lower layer SM estimates. The average change in weighted variance was -16%, -6%, and -
20% for estimates of SM5, SM6, and SM7, respectively.
**3.1.2. Impact on NDVI and crop yield**
Overall, in comparison to the free model, SDA improved yield estimates by explaining 17.7% more variation
in observed yield values and improving yield accuracy in 63% of site-years (Table 3). SDA accuracy was most
effective in site-years facing greater water stress. In those cases where yield estimates were improved, SDA often
increased available soil water at critical points in crop development, reducing crop soil water deficit factors and
increasing yield compared to the free model (Fig. A1). The most evident example of SDA yield improvement is IN in
2012, where the free model estimated complete maize crop failure (i.e., no grain yield) due to leaf senescence in mid-
July, but SDA estimated a harvestable crop due to increased soil water in the early season (Fig. 5). However, SDA's
impact on yield precision was inconsistent; roughly 53% of site-years saw reduced precision in yield estimates.
Overall, the free model accurately captured the phenological development of the cropping systems simulated
in this study, as demonstrated by the good agreement between observed and simulated NDVI (Fig. A2). SDA's impact
on NDVI accuracy was similar to its impact on yield accuracy, such that it typically either increased accuracy due to
lessened water stress or did not substantially affect the model performance. A comparison of R2 values demonstrates
that SDA helped to explain 4.8% more variation in observed NDVI values compared to the free model. Intuitively,
the site-years with the greatest jumps in NDVI accuracy also usually showed great improvement in yield accuracy,
highlighting a well-defined physiological relationship between vegetation and grain yield in APSIM's Maize and Plant
modules. SDA's impact on NDVI precision was inconsistent, such that 63% of site-years reduced precision in
estimates.
**3.1.3 Impact on tile drainage and nitrate load**
Across the 19 site-years, the free model and SDA showed overall poor performance in estimating annual
drainage with nRMSE values ranging from 18-215% with a median value of 54.3% for SDA and from 20-250% in
the free model with a median value of 52.4% (Fig. A4). In the site-years with the lowest accuracy, APSIM often
overpredicted drainage in both the free model and SDA. However, these cases of considerable overestimation in
drainage were also among those site-years that were most improved by SDA. 8 of the 11 site-years where SDA
improved estimates of annual drainage were cases where the free model overestimated tile flow. In these scenarios,
SDA functioned to remove available water from the soil profile and correctly lower the amount of water lost from the
system. In the remaining site-years where SDA did not improve drainage accuracy, SDA increased RMSE values by
32% on average. SDA's impact on precision for annual drainage estimates was highly variable. 63% of site-years saw


improvement in precision, but four site-years saw an immense reduction in precision (i.e., between 107-146%
reduction).
APSIM also struggled to accurately estimate the annual NO3 load for the tested site-years in this study (Fig. A3). For
the free model, nRMSE values ranged from 23-681% with a median value of 83.7% and, for SDA, nRMSE values
ranged from 17-833% with a median value of 86.9%. Considering the SDA constraint, estimates of annual NO3 load
were the most poorly constrained in terms of accuracy and precision. SDA's impact on precision was split, increasing
precision in 53% of site-years. Accuracy was improved for just 32% of site-years. Among those six site-years where
SDA increased NO3 load accuracy, SDA typically reduced estimates compared to the free model. Improved sites were
often maize years characterized by high input winter precipitation (Jan-Apr). No clear environmental nor agronomic
trend was identified among those 11 site-years where SDA reduced accuracy.

### 448    3.2  Assimilation of remote sensing soil moisture products

### 449    3.2.1 Individual assimilation runs

As expected, the individual influence of each RS data product was heavily dependent on its multi- or single-
sensor design and temporal availability. ESA, the most widely available data product, had the greatest impact on both
assimilation and downstream state variables. In contrast, assimilation with 1KM and 3KM imposed only slight
changes in estimates when compared to the free model. However, ESA did not always lead to improvements in model
performance. As demonstrated in Figure 6a, ESA results were more variable across site-years in terms of the accuracy
of state variable estimates, in some cases leading to great improvement and, in other cases, leading to reduced
performance. ESA reduced accuracy in predicting SM3 and SM4 in most site-years (i.e., 80-90%) but was the most
effective in improving accuracy in estimates of annual yield, SM6, and SM7. ESA also outperformed the other 3 RS
data products in constraining forecast precision for all state variables, improving precision in 70-100% of site-years.
Importantly, it showed the greatest reduction in the spectral norm of the SM covariance matrix when compared to the
free model, indicating the best constraint of SM precision across the entire profile (Fig. A7).
Alternatively, the assimilation of SMAP-HB, another temporally frequent RS data product, demonstrated
more conservative performance than ESA across state variables. For almost all state variables,
it also performed similarly or better than the free model. However, any improvements (or reductions) in forecast
accuracy were more moderate than observed with ESA. For example, accuracy in yield estimates was improved more
consistently with SMAP-HB (90%) compared to ESA (70%), but the maximum improvement in a tested site-year was
a 53% accuracy increase compared to a 95% increase with ESA. This trend in the results highlights an important trade-
off when assimilating more certain observations (i.e., ESA-CCI) at a coarse spatial resolution over less certain
observations at high spatial resolution (i.e., SMAP-HB) when both data products have unknown biases. In terms of
forecast precision, SMAP-HB was overall quite effective in constraining state variable predictions, especially when
compared to 1KM and 3KM. However, SMAP-HB underperformed compared to ESA in this regard. 1KM and 3KM
both underperformed in accuracy constraint when compared to ESA and SMAP-HB, showing little to no change in
RMSE compared to the free model.





Considering the four individual runs, more frequent assimilation time steps also led to a more robust
performance of the EnKF-Miyoshi workflow. Filter divergence (i.e., when the observed mean falls outside of the 95%
credibility interval of the analysis distribution) occurred at 52% and 59% of analysis time steps for 1KM and 3KM,
respectively, but occurred at only 44% and 30% of analysis time steps for SMAP-HB and ESA, respectively. For
estimates of observation uncertainty, the Miyoshi algorithm predicted greater uncertainty for most RS observations
than what is reported in the literature. The average standard error in ESA observations was reported to be $0.02 \pm 0.004$
mm3/mm3 but estimated in this study as $0.05 \pm 0.01$ mm3/mm3. Standard errors in 1KM and 3KM estimates were
reported as 0.05 m3/m3 but estimated by the system to be $0.07 \pm 0.02$ mm3/mm3 and $0.06 \pm 0.01$ mm3/mm3,
respectively. Miyoshi estimated similar uncertainty values for SMAP-HB observations as reported in the literature
(i.e., $0.07 \pm 0.02$ mm3/mm3).
**3.2.2 Additive runs**
The baseline run for the additive RS-SDA runs was ESA, which demonstrated inconsistent constraint of
forecast accuracy and strong constraint of forecast precision. The second most available data product, SMAP-HB, was
the next RS data product added to the system. New SMAP-HB observations, on average, imposed a -0.012 mm/mm
change in $\mu_a$ and a -0.0003 change in $P_a$ for SM1 estimates. For downstream forecast accuracy, the addition of SMAP-
HB led to improved and/or more consistent constraints for all state variables except SM7 (Fig. 6b). At times, the added
information from SMAP-HB dampened the benefit of SDA, reducing accuracy improvement. For forecast precision,
+SMAP-HB precision was overall better than the free model but with reduced performance compared to ESA.
The subsequent additions of the sparser 1KM and 3KM RS data products were less impactful than the
addition of SMAP-HB. New 1KM observations imposed an average -0.0004 mm/mm change in $\mu_a$, and, later, new
3KM observations imposed an average -0.0003 mm/mm change in $\mu_a$. These changes were less than 4% of the change
imposed by the initial addition of SMAP-HB. Neither additional data product produced a notable average change in
$P_a$. Following these minimal changes in SM1, there was also little change in forecast accuracy and precision for
downstream state variables in +1KM and ALL when compared to +SMAP-HB (Fig. 6b). Adding 1KM observations
to +SMAP-HB did hold some benefit for accuracy and precision in SM3 and SM4, while the effect of the 3KM
observations was almost negligible or, even at times, harmful to system performance.
**3.2.3 Impact on APSIM model estimates**
When considering the impact of surface SM data assimilation on downstream model variables, we focus on
results where all available RS observations were assimilated for each site . Hereinafter, we refer to the compilation of
these runs across the five sites as RS-SDA.
Overall, RS-SDA had minor impacts on the soil water profile relative to the free model. Figure 7 demonstrates
differences between the free model and RS-SDA in SM1 estimates. For several site-years, RS-SDA estimated
significantly higher SM1 values in the early growing season (i.e., May-Jun). In the late season and fall, RS-SDA often
estimated lower SM1 values. The impact of these SM1 changes on lower layer SM values seemed to decrease with
depth, such that differences between the free model and RS-SDA mean estimates were more subtle in deeper layers.
This reduced impact on lower layers is also, in part, a reflection of the increasing total soil water volume represented
by soil layers down through the profile (see Table 3 for layer depths). Nonetheless, any differences in SM estimates





did not lead to notable changes in accuracy for any SM layer (Table 3). Notable changes were visible in the soil water
deficit factors for several growing seasons, such that RS-SDA led to reduced water stress for the growing crop. We
speculate that this results from increased available soil water in the root zone during initial periods of crop water
uptake (i.e., June). Forecast precision for soil water-related estimates also did not change substantially with
assimilation. For SM1 estimates, assimilation substantially reduced variability across site-years (Fig. 7). In many
cases, this constraint in the surface soil layer did not propagate into significant changes for precision in lower layer
estimates (Fig. 6). However, on average, precision was improved rather than reduced with assimilation, with the most
significant downstream constraint in the soil layers closest to the surface.
RS-SDA demonstrated partial constraint of aboveground estimates. Considering the R2 values reported in
Table 3, RS-SDA explained roughly 4% more variation in yield observations than the free model. All site-years except
OH 2015 demonstrated increased yield accuracy, and 60% of sites demonstrated increased yield precision with RS-
SDA. Based on these results, there is evidence that surface SM data assimilation can constrain, to some extent,
estimates of annual yield. There was no significant relationship between yield improvement and dry conditions, though
this could be an artifact of sample size (Fig. A4). Compared to its effect on yield estimates, RS-SDA was less impactful
in its constraint of NDVI. However, since the free model could reasonably predict NDVI (R2 = 0.69), there was less
potential for improvement with SM assimilation. 60% of site-years had increased accuracy, and 70% had increased
precision for NDVI estimates following SDA.
**4. Discussion**
**4.1 Sensitivity of APSIM model estimates to in situ soil moisture**
In this study, the extent to which in situ SM data assimilation affected APSIM model predictions depended
on each state variable's sensitivity to the assimilated state variable (i.e., soil moisture). Deeper layer SM estimates—
the most sensitive state variables to SM3 and SM4—were the most strongly constrained. Figure A1 demonstrates the
significant linear relationship between daily changes in forecasted SM3 and SM4 due to SDA and daily changes in
SM estimates for all deeper soil layers. As expected with a cascading water balance model, the strength of the linear
relationship weakens as the vertical distance between soil layers increases. In the model, SM in each layer can
influence SM estimates of deeper soil layers, but only indirectly through its influence on the SM in the layer
immediately below it. Therefore, the influence of the assimilation layers is reduced by each subsequent SM process
down through the soil profile and is weakest in the final soil layer (SM7). Nevertheless, the constraint of SM7 was
still quite strong in SDA. By assimilating SM for two upper soil layers, the accuracy of SM estimates improved
immensely by simply leveraging the pre-existing model structure (compare to Liu et al., 2017).
Crop yield showed the next strongest constraint in SDA. However, as noted in previous studies, its sensitivity
to SM SDA was conditional (Lu et al., 2021; Kivi et al., 2022). While changes in SM affected lower layer SM at all
analysis time steps, crop yield was only affected when the changes impacted crop water stress. Daily crop water uptake
is determined in APSIM as the minimum of crop water demand and soil water supply. Therefore, SDA could only
influence crop yield when the soil water adjustment pushed the water supply above or below the demand threshold.





For this reason, greater SDA improvement was found in crop yield estimates during water-stressed site-years. Other
pathways through which SM can impact crop yield in APSIM, like soil N cycling, did not play a strong role in this
study.

The impact of SM SDA on APSIM drainage estimates can also be beneficial given certain conditions. As

shown in the results, drainage was affected by SM3 and SM4 through 2 pathways: (1) changes in total soil water with
assimilation adjustment and (2) changes in crop water uptake due to changes in crop water stress. The role of each of
these pathways varied over the year, such that the presence of a growing crop and root system weakened the sensitivity
of drainage estimates to changes in the assimilation layers. To quantify this change in sensitivity, we divided daily
model forecasts into two categories: with crop water uptake (June-Sept) and without crop water uptake. Then, the
relationship between changes in SM3 and SM4 and changes in drainage was analyzed separately for each group. There
was no significant linear relationship when looking at SM3 changes in either case. However, the linear relationship
between changes in SM4 and changes in daily drainage was stronger when no crop was present ($r = 0.23$, $p = 0.00$)
than when a crop was present ($r = 0.14$, $p = 0.00$). This is similar to Hu et al. (2008), who identified notable changes
in drainage dynamics during rapid crop growth compared to out-of-season dynamics in SPWS model simulations.

Among the state variables considered in SDA, NO3 leaching showed the weakest and most complex

relationship with SM3 and SM4 in APSIM. Therefore, logically, the presented system performed most poorly in its
constraint of annual NO3 leaching estimates. In APSIM, daily NO3 leaching estimates are computed as the product
of two different daily values: estimated NO3 concentration in the lowest soil layer and estimated tile drainage.
Therefore, in addition to its impact on drainage, SDA can affect NO3 load estimates through (1) changes in N cycle
processes via SM rate factors (see Fig. 2 in Kivi et al., 2022) and (2) changes in the vertical movement of soil water
(and N solutes) through the soil profile. In a validation study of APSIM N processes, Sharp et al. (2011) also observed
inconsistent model behavior in annual leaching estimates for their experimental site in New Zealand when simulating
three years of a potato-rye rotation. Their final calibration of the model only improved one of the annual estimates but
did not constrain estimates in the other two years. In fact, many past studies have highlighted nitrate leaching estimates
as a broader forecasting challenge (Stewart et al., 2006; Sharp et al., 2011; van der Laan et al., 2014; Brilli et al.,
2017). As highlighted already in the literature, missing processes related to snowmelt (Ojeda et al., 2018), and tillage-
related infiltration (Malone et al., 2007; Brilli et al., 2017; Ojeda et al., 2018), or preferential flow could help to
improve APSIM performance. Though there is still potential for the presented system to improve nitrate leaching
estimates, further investigation and constraint of the APSIM N and soil water cycles will be necessary to ensure
consistent performance.
**4.2 Impact of remote sensing soil moisture data assimilation**

The assimilation of RS surface SM observations imposed a far weaker constraint on APSIM state variables

compared to the assimilation of the soil sensor observations. For example, the median reduction in SM RMSE ranged
from 7-27% across different layers of the soil profile with soil sensor observations, but, with RS observations in RS-
SDA, it ranged from roughly 1-5% (Table 3). The weakened constraint with RS-SDA was likely more than an issue
of observation inaccuracies. Instead, there is greater evidence to show that changes in SM1 simply had less influence


on downstream state variables than changes in SM3 and SM4. This is due, in part, to the increased vertical distance
between the surface SM layer (SM1) and other observed soil layers (i.e., SM3-7). The APSIM SoilWat module
operates as a cascading water balance model to estimate the movement of water and solutes between and across soil
layers (Dokoohaki et al., 2018). Thus, the assimilation adjustment of the SM1 estimate would not be as strongly tied
to lower layer estimates when using a top-down approach. Yet, surface SM data assimilation notably changed SM2
estimates, the SM estimates for the layer just below it. This result reflects the findings of Lu and Steele-Dunne (2019),
who assimilated RS surface SM observations into a surface energy balance model. They found that SDA improved
SM estimates in the second layer to a greater extent than in lower layers when comparing estimates to observations.
Since observations were not available for SM2 at the study sites, this hypothesis could not be tested within this work.

The two assimilation protocols (i.e., assimilation of SM1 vs. assimilation of SM3 and SM4) were also

markedly different in the quantity of soil water associated with their assimilation adjustments. Where soil layers 3 and
4 corresponded to almost 14% of the soil profile (20 cm depth), the near-surface soil layer only corresponded to about
3.6% of the soil profile (5 cm depth). Thus, when considering the top-down effect of SM assimilation on lower layers,
each adjustment with RS assimilation had just 25% of the impact of the previous system given the same adjustment
in volumetric soil water content. This 5-fold reduction in potential impact closely mirrors the change in RMSE
reduction for SM layers highlighted above (i.e., 7-27% to 1-5%). One way to overcome this limitation of surface SM
is to leverage the strong covariance between SM1 and SM in nearby layers (i.e., SM2) to directly nudge their values
within the analysis time step using, for example, an augmented state vector (e.g., Kivi et al., 2022) or exponential filter
approaches (e.g., Albergel et al., 2008).

RS surface SM data assimilation still demonstrated strong potential for improving APSIM forecasts within

this study. First, the assimilation of surface SM improved estimates of crop yield overall when compared to the free
model, with a median RMSE reduction of 17.2%. Past RS SM data assimilation studies had similar success in
improving crop yield estimates, and several attributed the improvement to increased surface SM and reduced crop
water stress with SM assimilation (e.g., Ines et al., 2013; Chakrabati et al., 2014). We speculate that the model
performance indicate that water stress likely played an important role. Although direct observations are not available
for crop water uptake to test this hypothesis, we suspect RS-SDA accurately increased available soil water at critical
growth stages and, thus, increased crop water uptake.
**4.3 Comparison of remote sensing soil moisture data products**

The four different RS SM data products varied quite broadly in spatial resolution, varying from 30 meters to

0.25°. However, their individual assimilation performance seemed to be most closely tied to the temporal availability
of observations.  ESA with a multi-sensor nature had an average, 219 observations per growing season and showed
the best overall constraint of forecast precision and good constraint of forecast accuracy in downstream state variables.
Alternatively, the 1KM and 3KM data products, which each had an average of 7 observations per growing season,
had almost no impact on forecast accuracy and only a slight impact on forecast precision. Although this study was not
designed to independently test the impact of temporal and spatial resolution on performance, it echoes the findings of



Lu et al. (2019), who found a high temporal resolution to be far more important to assimilation performance than high
spatial resolution. They suspected that increased time between assimilation adjustments allowed errors in model
structure, inputs, and/or parameters to go unchecked for more extended periods of time, thereby allowing the
magnitude of simulation errors to become large and unreasonable. More frequent assimilation helps mitigate the
impact of such model errors and improve overall crop model predictions by correcting errors more often (De Lannoy
et al., 2007; Pauwels et al., 2007; Lu et al., 2021). Alternatively, in the case of low temporal resolution, a recalibration-
based assimilation approach or the inclusion of a bias correction method might be more appropriate (De Lannoy et
al., 2007; Curnel et al., 2011).

When comparing RS data products in this study, it is important to recognize that all data products considered

in this work are based, in part, on SMAP radiometer data. SMAP-HB merged SMAP brightness temperature data with
the HydroBlocks-RTM model, ESA includes SMAP as one of its ten passive microwave sensors, and 1KM and 3KM
rely on SMAP for passive microwave information within their derivation. In the first iteration, ESA contributed most
of the information provided by the SMAP radiometer to the model and, therefore, imposed large changes in SM1
estimates. Then, with each additional data product, the overall impact on the analysis distribution weakened as much
of the new information had already been provided to the system.

The Miyoshi algorithm often estimated higher observation uncertainty (R) than the values reported in the

literature. This is unsurprising as RS SM data products, like most RS data products, often have poorly characterized
uncertainties (Peng et al., 2021). For each data product, uncertainty is typically reported as a standard error value after
comparing the data product to a limited set of observations. This estimate does not capture all possible sources of
uncertainty and cannot be easily generalized to different places or time points (Huang et al., 2019). Yet, in the additive
runs, these uncertainty values were applied uniformly across time and space. Future applications of the GEF scheme
could benefit from additional terms in the model that could capture R or the use of the Miyoshi algorithm. These
approaches may better estimate observation uncertainties within the system's context.

## 5. Conclusions

In the study, we assessed the extent to which soil moisture data assimilation can improve APSIM model forecasts.
Building on Kivi et al., (2022), we used a generalizable and novel data-assimilation system to assimilate RS and in
situ soil moisture measurements across the U.S. Midwest 19 site-years, and evaluated how direct soil moisture
constraint affected downstream model estimates of root-zone soil moisture, crop yield, tile flow, and nitrate leaching.
Our results highlighted the capacity of soil moisture data assimilation to improve model estimates of crop yield in
water-limited conditions, increasing crop water uptake at critical points in the growing season. Soil moisture data
assimilation also improved estimates of soil moisture throughout the profile in most cases but did not well constrain
nitrate leaching or tile drainage. This indicates a need for better constraint of both the soil water and soil nitrogen
cycles in the APSIM model.
This work also lays the groundwork for future regional applications of soil moisture data assimilation. Importantly,
our findings reaffirmed soil moisture data assimilation's ability to "localize" gridded weather estimates of precipitation
to reflect observed values more accurately. Since cropping systems are highly sensitive to precipitation inputs, this is



a strong advantage of soil moisture data assimilation for forecasting applications where coarse-resolution weather drivers are employed. Though RS soil moisture data assimilation could be an effective way to overcome limited availability of in situ data, our work shows that assimilation of in situ surface soil moisture is not as powerful as the assimilation of in situ root-zone soil moisture values in terms of model constraint. If the former is applied, additional constraints or an augmented state-vector approach would be necessary to achieve higher system performance. When selecting a RS soil moisture data product for data assimilation applications, high temporal resolution due to multi-sensor satellite availability and accurately estimated observation uncertainty are two critical components for optimal system performance. To that same point, combining several data products at different spatial resolutions can help to reduce assimilation intervals within the system. Further investigation is needed to independently test the impact of observation sample size (i.e., number of data products), temporal resolution, spatial resolution, and uncertainty on system performance. Moreover, the data products considered in this study do not represent the full range of RS soil moisture data products that are available publicly. This work should be expanded to evaluate data products derived from other satellites/derivations both individually and in combination with other sources to exhaust all available options.

**6. Code and data availability**

Code and observational data used in this study will be provided upon request.

**7. Author contribution**

MK was responsible for code development, performing the simulations and writing the manucript. NV contributed to revising the manuscript and providing SMAP-HB dataset. HD was responsible for developing the initial idea, code development, writing and supervising the study.

**8. Competing interests**

The contact author has declared that neither they nor their co-authors have any competing interests.

**9. Acknowledgements**

The authors would like to thank all those on the Energy Farm team who made the presented case study possible. In particular, we would like to thank Carl Bernacchi, Bethany Blakely, Michael Masters, Grace Andrews and Heather Goring-Harford, who made the Energy Farm dataset available and performed the analyses for the nitrate leaching data, and Konrad Taube and Haley Ware, who helped with water collection and water filtering in 2018 and 2019. We also want to thank Caitlin Moore and Evan Dracup, who helped to collect and process much of the other data from the plot. Additionally, we wanted to acknowledge those funding sources that supported the work of the Energy Farm team. First, the data used in this study was funded in part by (1) the Leverhulme Centre for Climate Change Mitigation,



funded by the Leverhulme Trust through a Research Centre award (RC-2015-029), (2) the Center for Advanced Bioenergy and Bioproducts Innovation (CABBI) at the University of Illinois, and (3) the Global Change and Photosynthesis Research Unit of the USDA Agricultural Research Service.

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



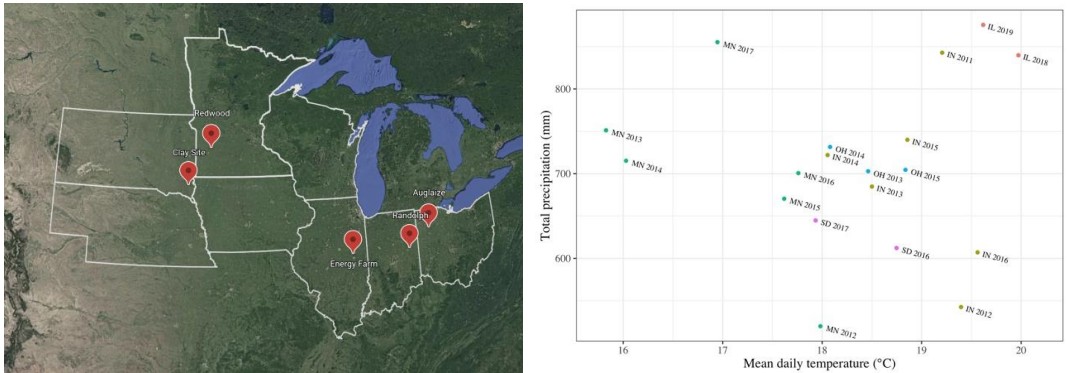

**Figure 1. (A) Site map (ESRI) and (B) scatterplot demonstrating site-year total precipitation and average daily temperature (°C) for each site-year between April and October. Climate information was extracted and averaged across the 10 ERA5 weather ensembles for each site-year.**


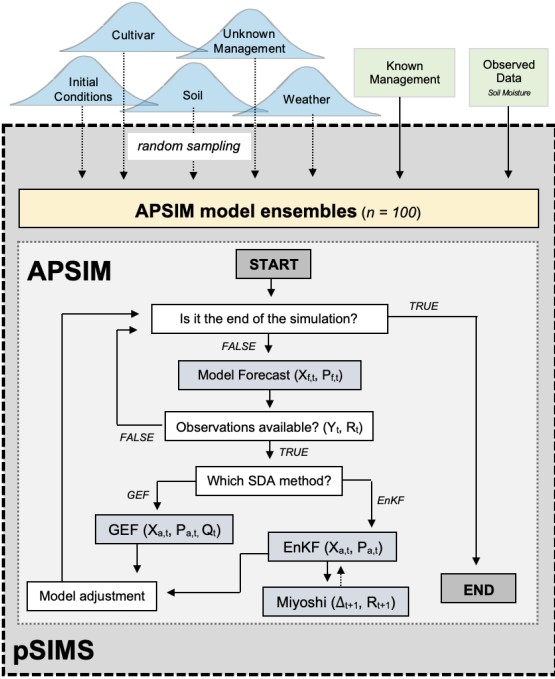

**Figure 2. Schematic demonstrating the workflow of the data assimilation system. System inputs represented by blue Normal distributions have incorporated uncertainty in this study, while green rectangles represent known values that were included as constants.**


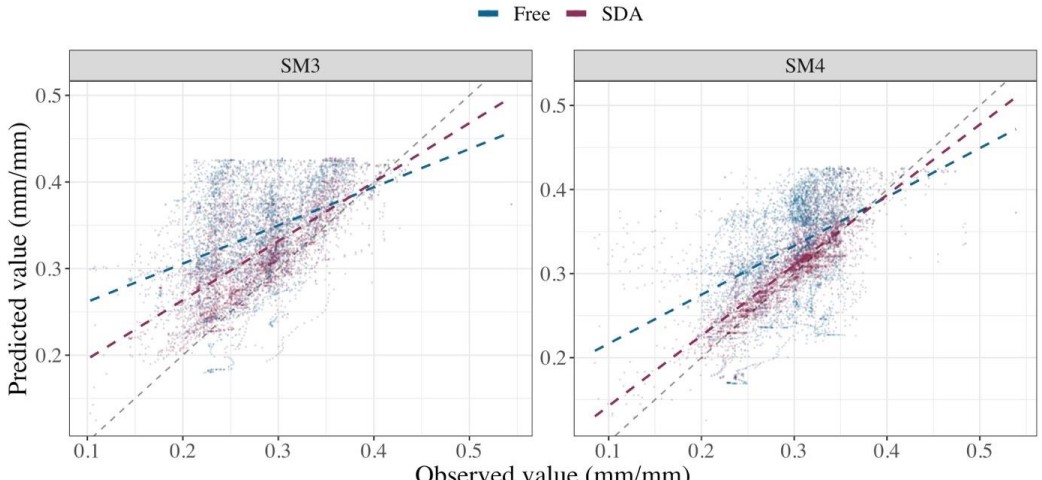

**Figure 3. One-to-one plots for soil moisture estimates (mm/mm) in the two assimilation layers for the free model and in situ SDA across all analysis time-steps and site-years. The least-squares regression line is shown for both schemes next to the black dashed 1:1 line, demonstrating a perfect fit.**


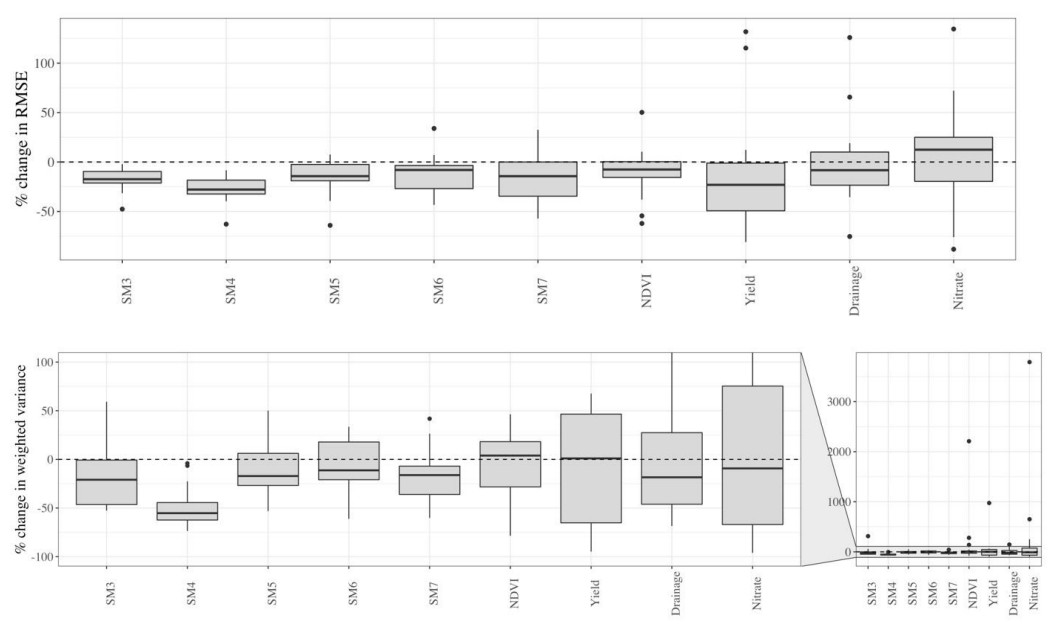

**Figure 4. Boxplots demonstrating the distribution of relative change in (a) accuracy (RMSE) and (b) precision (weighted variance) due to in situ SDA for each state variable across all site-years. The relative change is computed with respect to the free model run, with negative values indicating SDA improvement.**




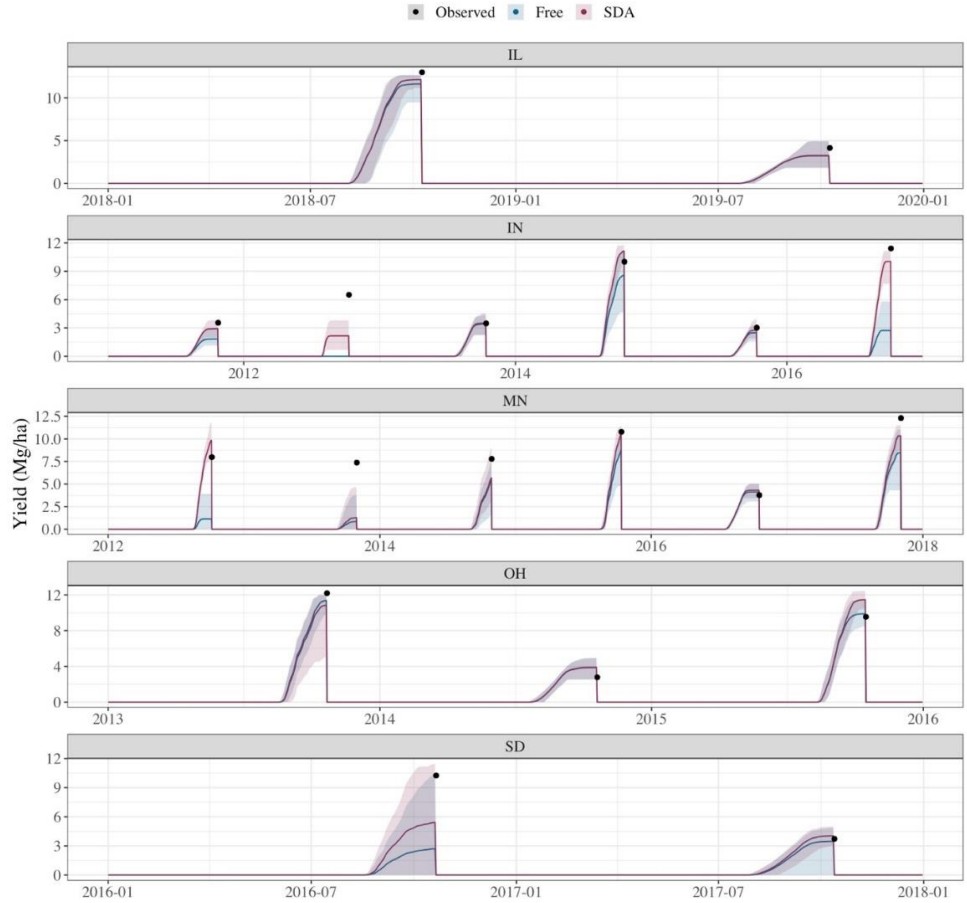

**Figure 5. Time series of yield estimates for the free model and in situ SDA with mean daily estimates demonstrated with line graphs and the 95% credible intervals demonstrated by the shaded regions. Black points represent the observed harvest date and yield for each site-year.**




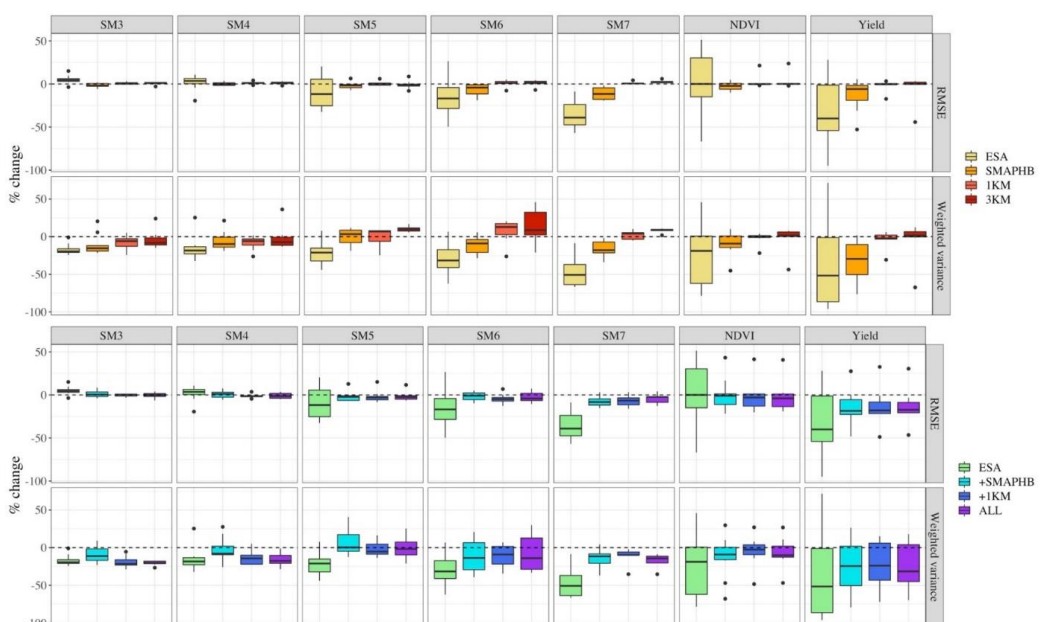

**Figure 6. Boxplots demonstrating the distribution of relative change (%) in state variable accuracy (RMSE) and precision (weighted variance) for the (a) individual and (b) additive runs across all site-years. Change is computed relative to the free model results. Negative values indicate improvement (e.g., (RMSES – RMSEF) / RMSEF).**



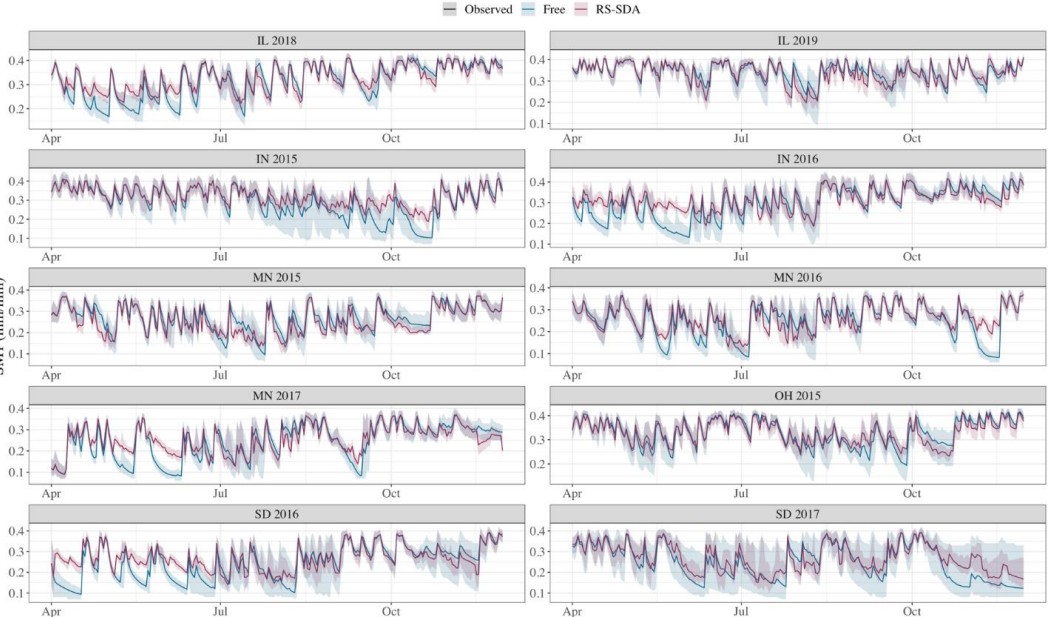



**Figure 7. Time series of SM1 estimates from the free model and RS-SDA with the mean daily estimates demonstrated with line graphs. The shaded regions indicate 95% credibility intervals.**


**Table 1. Overview of remote sensing soil moisture data products.**

| Product | Product ID | Temporal coverage | Temporal frequency | Spatial resolution | Average data availability | Average observation variance | Reference |
|---|---|---|---|---|---|---|---|
| ESA-CCI | ESA | 1978-2019 | 1-2 days | 0.25° | 219 days | 0.0003 | Dorigo et al. (2017) |
| SMAP-Hydroblocks | SMAP-HB | 2015-2019 | 1-3 days | 30 m | 127 days | 0.0050 | Vergopolan et al. (2021b) |
| SMAP-Sentinel1 | 1KM/3KM | 2015-now | 12 days | 1 km/3 km | 7 days | 0.0025 | Das et al. (2019) |
| [a]Availability is calculated after removing observations in the winter months (i.e., Dec-Mar) and is given on a per-year basis. | | | | | | | |



**Table 2. Overview of system configuration for the nine runs performed in this study. SDA methods include the Ensemble Kalman Filter (EnKF) coupled with the Miyoshi algorithm, and the Generalized Ensemble Filter (GEF). The former method of these two methods provided systematic estimates of R applied within the system, but the latter method used literature values. The state variables included in Xf are given.**

| Run group | Name | SDA method | R estimates | Temporal extent | State variable(s) | Observation(s) |
|---|---|---|---|---|---|---|
| Baseline | Free | N/A | N/A | 2011-2019 | N/A | N/A |
| | SDA | EnKF | Miyoshi | 2011-2019 | SM3, SM4 | In situ soil sensor |
| Individual Runs | ESA | EnKF | Miyoshi | 2015-2019 | SM1 | ESA |
| | SMAP-HB | EnKF | Miyoshi | 2015-2019 | SM1 | SMAP-HB |
| | 1KM[a] | EnKF | Miyoshi | 2015-2019 | SM1 | 1KM |
| | 3KM[a] | EnKF | Miyoshi | 2015-2019 | SM1 | 3KM |
| Additive Runs | +SMAHB | GEF | Literature | 2015-2019 | SM1 | ESA, SMAP-HB |
| | +1KM[a] | GEF | Literature | 2015-2019 | SM1 | ESA, SMAP-HB, 1KM |
| | ALL[a] | GEF | Literature | 2015-2019 | SM1 | ESA, SMAP-HB, 1KM, 3KM |
| [a] Observations for 1KM and 3KM were not available for IL, and thus simulations were not performed for the site. | | | | | | |







**Table 3. Summary statistics to quantify the impact of in situ SDA (IS) and RS-SDA (RS) on forecast accuracy of APSIM state variables. The "Ns" column indicates the number of site-years with available data for each state variable and each run, and the "ns" column indicates the total number of observations across site-years for each run. A subscript (F) denotes a value computed for the free model estimates, a subscript (IS) denotes a value for the in-situ SDA estimates, and a subscript (RS) denotes a value for RS-SDA runs. The median change (D) in RMSE was computed for both runs. Two values for R2F are given for the different data subsets demonstrated in the "N" and "n" columns.**

| State variable | Depth (cm) | $N_{IS}$ ($N_{RS}$) | $n_{IS}$ ($n_{RS}$) | $\Delta$ $RMSE_{IS}$ | $\Delta RMSE_{RS}$ | $R^2_F$ | $R^2_{IS}$ | $R^2_{RS}$ |
|---|---|---|---|---|---|---|---|---|
| SM3 *mm/mm* | 9.1 – 16.6 | 19 (10) | 12252 (5592) | -17.4% | -0.9% | 0.49 (0.48) | 0.57 | 0.48 |
| SM4 *mm/mm* | 16.6 – 28.9 | 19 (10) | 12735 (6141) | -27.9% | -2.8% | 0.52 (0.43) | 0.73 | 0.43 |
| SM5 *mm/mm* | 28.9 – 49.3 | 17 (8) | 11325 (5101) | -14.3% | -2.6% | 0.45 (0.45) | 0.38 | 0.45 |
| SM6 *mm/mm* | 49.3 – 82.9 | 19 (10) | 12846 (6169) | -8.0% | -1.0% | 0.42 (0.43) | 0.34 | 0.42 |
| SM7 *mm/mm* | 82.9 – 138 | 9 (6) | 5715 (3265) | -14.3% | -5.4% | 0.43 (0.44) | 0.34 | 0.43 |
| NDVI *unitless* | - | 19 (10) | 244 (134) | -7.6% | -1.8% | 0.62 (0.69) | 0.66 | 0.71 |
| Yield *Mg/ha* | - | 19 (10) | 19 (10) | -23.1% | -17.2% | 0.55 (0.53) | 0.73 | 0.59 |
| Annual drainage *mm* | - | 19 | 19 | -8.3% | - | 0.47 | 0.46 | - |
| Annual NO₃ load *Kg NO₃-N/ha* | - | 19 | 19 | +12.5% | - | 0.42 | 0.45 | - |
