# Peer review of "A comprehensive assessment of in situ and remote sensing soil moisture data assimilation in the APSIM model for improving agricultural forecasting across the U.S. Midwest"

_Hydrology and Earth System Sciences, 2022_

## Referee Comment (RC2)

**General comments**

The study **"A comprehensive assessment of in situ and remote sensing soil moisture data assimilation in the APSIM model for improving agricultural forecasting across the U.S. Midwest"**, M.Kivi, N.Vergopolan, H.Dokoohaki comprises of a set of new approaches and techniques of data assimilation for improving agricultural forecasting. In this study authors integrated in situ and remote sensing soil moisture observations with APSIM model through sequential data assimilation and assessed the extent to which soil moisture data assimilation can improve APSIM model forecasts. Therefore, paper addresses relevant scientific questions within the scope of HESS. The scientific methods and assumptions present are valid. The title clearly reflects the contents of the paper and abstract provides a complete summary of the research.

The work showed that assimilation of in situ surface soil moisture is not as powerful as the assimilation of in situ root-zone soil moisture in terms of model constraint. It is shown that high temporal resolution due to multisensor satellite availability and accurately estimated observation uncertainty are critical components for optimal system performance. More frequent assimilation helps mitigate the impact of such model errors and improve overall crop model predictions by correcting errors more often. Assimilating in situ observations, the accuracy of soil moisture forecasts in the assimilation layers was improved by an average of 17% for 10 cm and ~28% for 20 cm depth soil layer across all site-years and the crop yield was improved by an average of 23%.

**Specific comments**

There are a number of questions:

- In this study, APSIM's daily forecasts of agricultural variables were transformed and used as inputs into the PROSAIL model to compute the spectral reflectance. Would it be the source of errors in future predictions?
- There are a number of crops used in the study, which have different spectral signatures, biomass, stages of development, nutrient uptake, water use and water stress effect etc. Therefore, in order to reduce errors of agricultural forecasting would it be better to use different optimized variables for each crop?

Some references should be revised in text and in the list of References

---

## Author Comment (AC1)

**Authors' Response to Reviews of**

**A comprehensive assessment of in situ and remote sensing soil moisture data assimilation in the APSIM model for improving agricultural forecasting across the U.S. Midwest**

Marissa Kivi, Noemi Vergopolan, and Hamze Dokoohaki
*Hydrology and Earth System Sciences (HESS),*
* * *
**RC:** *Reviewers' Comment*,     AR: Authors' Response,     ☐ Manuscript Text

**1. Reviewer #1**

**RC:** *Kivi et al (2022) is cited 25 times in the text! While this work is distinctly different to that work, perhaps a single mention at the start of each section, rather than every second and third level sub-section, would be adequate (such as the existing introductory paragraph in each major section).*

AR: Thank you for the comment. As per your suggestions, we reduced the number of citations to the Kivi et al., (2022) to only cases where it was necessary in describing a method or making a comparison with earlier works.

**RC:** *As far as I can tell, and I am no expert in data assimilation techniques or applications, the work presented is solid and thoroughly describes the proper use of the techniques. The authors honestly presented when the technique improved certain predictions, when it made little difference, and when the model performance degraded. While I can see why the extension was required in Figure 4b, as it's only 3 of the results and the big picture is shown well with the box-and-whiskers, the extension can be placed in the Appendices and mentioned in the text as with free-model results.*

AR: Thank you for the comment. Since constraining uncertainties is a central idea to the application of sequential data assimilation (SDA) methods -as important as reducing model biases- we initially decided to present the both metrics to show the emphasize of this feature in SDA. If possible we would like to keep the figure in the way in which it is presented . If the reviewer believes that it is necessary to make the change, we will be happy to do so.

**RC:** *Was the "free run" calibrated against any data, or was it just an ensemble of model runs with the parameters randomly assigned from prior distributions? Were they a single run with an arbitrary (or literature) set of values assigned to parameters? How do the modelled results compare to the "final" set of parameters after the full SDA? I did not get the appendices so don't know if this is covered.*

AR: Thank you for the comment. Model parameters for describing crop growth and development was adopted from Dokoohaki et al. (2022), while other priors such as initial soil water content were set based on a plausible range for each variable (presented in detail in Table A.2 Kivi et al., (2022)). Within the SDA framework, no parameter was directly adjusted/optimized and main role of SDA is to adjust model state variables such as soil moisture rather than model parameters. SDA attempts to nudge the model prediction of the state variables towards observation (remote sensing or sensor) and in an ideal scenario this helps with improving system representation (which can be assessed through evaluating other state variables such as crop yield, NDVI, drainage and etc as performed in this study).

Dokoohaki, H., Rai, T., Kivi, M., Lewis, P., Gómez-Dans, J.L. and Yin, F., 2022. Linking Remote Sensing with APSIM through Emulation and Bayesian Optimization to Improve Yield Prediction. Remote Sensing, 14(21), p.5389.

Kivi, M.S., Blakely, B., Masters, M., Bernacchi, C.J., Miguez, F.E. and Dokoohaki, H., 2022. Development of a data-assimilation system to forecast agricultural systems: A case study of constraining soil water and soil nitrogen dynamics in the APSIM model. Science of The Total Environment, 820, p.153192.

There are a number of questions that the work raises, that may be answered here (or later maybe).

**RC:** *As the parameters are nudged successively with the SDA procedure, how much do they need to vary until the researcher considers that (a) their a priori range is incorrect, (b) that they cannot be considered a constant, or (c) that some process is too simplified or missing in the numerical model?*

Thank you for the comment. One main difference between SDA and classical calibration methods is that SDA nudges the model state variables and not the model parameters. But a model calibration is a complementary step that could be taken in addition to the SDA. For calibrating crop parameters in this study, we used regional model parameters provided by Dokoohaki et al., (2022) and then focused on quantifying the information contribution of different remote sensing data products in this study compared to gold standard sensor SDA. Overall this study attempts to understand the value provided by soil moisture remote sensing data products for improving the prediction of other processes (drainage, N leaching, crop yield and etc) where soil sensor is not readily available. Furthermore, one of the benefits of SDA is to actually correct the model state errors due to wrong parameterization and/or lack of process representation.

Dokoohaki, H., Rai, T., Kivi, M., Lewis, P., Gómez-Dans, J.L. and Yin, F., 2022. Linking Remote Sensing with APSIM through Emulation and Bayesian Optimization to Improve Yield Prediction. Remote Sensing, 14(21), p.5389.

**RC:** *Further to the case of a non-constant model parameter, is there a pattern in the parameter adjustments, e.g., always too high in winter and too low in summer, that indicates a systematic misrepresentation within the model? Is there a form of a posteriori distribution of parameter values, e.g., normal versus log-normal versus bi-modal versus uniform, that may indicate systematic model or data errors?*

**AR:** Although there is the possibly of including model parameters in the SDA in addition to the model state variables, we didn't explore this avenue in this study. This was mainly due to the fact that in SDA we only have a single observation for nudging model state variables and including model parameters might have resulted in over-fitting (nudging many unknowns with only one uncertain remote sensing observation). However in Kivi et al., (2022) we explored this avenue with more confident soil sensor observations and we found that adjusting the SWCON model parameter (a parameter controlling water flow between soil layers) for the two assimilation layers, though marginally helpful, did not dramatically improve soil moisture estimates as compared with no change in parameters.

**RC:** *Is there parameter bias (or trends?) associated with larger underlying groupings, such as soil texture, vertical layering (duplex, gradational, uniform), crop type, or management, that indicate model structure or data limitations?*

**AR:** This is a very interesting hypothesis. Even though we did not explore change in the parameters, we believe potentially using a hierarchical data assimilation framework and in a augmented system representation (states + parameters), there is a possibly of including site effects in the statistical model and exploring the relationship between the site effects and different environmental factors. However, the main limitation for this idea is the availability of high quality soil moisture data for a hierarchical data assimilation scheme.

**RC:** *Given the use of numerical models is primarily predictive, i.e., what are future potential grain yields or nitrate loss or deep drainage under different management or climate conditions, which set of parameters (or reduced range) do researchers consider stable enough to make such computations?*

**AR:** Thank you for the comment. SDA can improve initialization conditions for forecast runs, such that, although we do not have future remote sensing observations, by assimilating the observations we know that we have a well represented the initial conditions needed for runs of short term (next days, weeks) forecasts.

**RC:** *The References and citations need a lot of purely technical corrections.*

**AR:** All following technical comments were addressed.

**RC:** *Following reference not cited in text: Akhavizadegan et al (2021); Archontoulis et al (2014, 2020); Balboa et al (2019); Crane-Droesch (2018); Das et al (2020); Dietze et al (2013); Dietzel et al (2016); Flathers and Gessler (2018); Guerif and Duke (2000); Hoffman et al (2020); Jeong et al (2016); Kang et al (2020); van Klompernburg et al (2020); Leng and Hall (2020); Li et al (2014); Martinez-Feria et al (2019); Pasley et al (2021); Puntel et al (2016); Shahhosseini et al (2021); Spijker et al (2021); Wallach et al (2021).*

**AR:** Thank you for the comment. The reference section was accordingly adjusted to reflect only cited references in the text.

**RC:** *Following citations not in references: Lu et al (2019); Lu and Steele-Dunne (2019).*

**AR:** Thank you for pointing this out, this error was addressed.

**RC:** *The two articles of Vergopolan et al (2021) are cited with (a) and (b) in the text, but not indicated as such in the references.*

**AR:** Thank you for the comment. This error was fixed in the references.

**RC:** *Citation for Chakrabarti et al (2014) is misspelled in the text line 83 and 605.*

**AR:** Thank you for the comment. This error was fixed.

---

## Author Comment (AC2)

**Authors' Response to Reviews of**

**A comprehensive assessment of in situ and remote sensing soil moisture data assimilation in the APSIM model for improving agricultural forecasting across the U.S. Midwest**

Marissa Kivi, Noemi Vergopolan, and Hamze Dokoohaki
*Hydrology and Earth System Sciences (HESS),*

RC: *Reviewers' Comment*,     AR: Authors' Response,     ☐ Manuscript Text

**1.  Reviewer #2**

RC:  *General comments*

The study "A comprehensive assessment of in situ and remote sensing soil moisture data assimilation in the APSIM model for improving agricultural forecasting across the U.S. Midwest", M.Kivi, N.Vergopolan, H.Dokoohaki comprises of a set of new approaches and techniques of data assimilation for improving agricultural forecasting. In this study authors integrated in situ and remote sensing soil moisture observations with APSIM model through sequential data assimilation and assessed the extent to which soil moisture data assimilation can improve APSIM model forecasts. Therefore, paper addresses relevant scientific questions within the scope of HESS. The scientific methods and assumptions present are valid. The title clearly reflects the contents of the paper and abstract provides a complete summary of the research.

The work showed that assimilation of in situ surface soil moisture is not as powerful as the assimilation of in situ root-zone soil moisture in terms of model constraint. It is shown that high temporal resolution due to multisensor satellite availability and accurately estimated observation uncertainty are critical components for optimal system performance. More frequent assimilation helps mitigate the impact of such model errors and improve overall crop model predictions by correcting errors more often. Assimilating in situ observations, the accuracy of soil moisture forecasts in the assimilation layers was improved by an average of 17% for 10 cm and  28% for 20 cm depth soil layer across all site-years and the crop yield was improved by an average of 23%.

Specific comments There are a number of questions:

RC:  *In this study, APSIM's daily forecasts of agricultural variables were transformed and used as inputs into the PROSAIL model to compute the spectral reflectance. Would it be the source of errors in future predictions?*

AR:  We appreciate your comment. On a daily basis, the APSIM model passes soil moisture and Leaf Area Index (LAI) measurements to the PROSAIL model in order to simulate canopy reflectance and derive spectral indices. While it is true that inaccurate estimation of LAI and soil moisture can lead to inaccurate estimates of spectral indices, we utilized this to assess the improvement of spectral indices through the assimilation of soil moisture. Furthermore, the soil moisture error in APSIM model is included in the Kalman gain computation, such that assimilation accounts for errors in the model when computing the Kalman gain (how much we will let our satellite observations update the model).

RC:  *There are a number of crops used in the study, which have different spectral signatures, biomass, stages of*

*development, nutrient uptake, water use and water stress effect etc. Therefore, in order to reduce errors of agricultural forecasting would it be better to use different optimized variables for each crop?*

AR: Thank you for your question. The APSIM model accounts for the differences in crop growth and development between corn and soybean explicitly through the use of two different crop growth models. The maize module, developed from a combination of the CM-KEN (Keating et al., 1991, 1992) and CM-SAT (Carberry et al., 1989; Carberry and Abrecht, 1991) models of maize (both derivatives of CERES-Maize, Jones and Kiniry, 1986) with some features of the maize model of Wilson et al. (1995), is used to describe the growth and development of maize in APSIM. For soybean, the generic plant module is used, which currently includes crops such as chickpea, mungbean, cowpea, soybean, pigeonpea, stylosanthes, peanut, faba bean, lucerne, canola, weed, mucuna, lupin, wheat, and navybean. Also the SDA at each site was performed independently, such that the system is already designed for optimal assimilation considering each site crop-specific characteristics.

- Carberry, P.S.; McCown, R.L.; Muchow, R.C.; Dimes, J.P.; Probert, M.E.; Poulton, P.L. and Dalgliesh, N.P. 1996b. Simulation of a legume ley farming system in northern Australia using the Agricultural Production Systems Simulator. Aust. J. Exptl. Agric. 36: 1037-48.

- Keating, B. A.; Godwin, D. C.; Watiki, J. M. 1991. Optimising nitrogen inputs in response to climatic risk. In: RC Muchow and JA Bellamy (Eds) Climatic risk in crop production: Models and management in the semi-arid tropics and sub-tropics. Cab International, Wallingford . P. 329-358.

- Keating, B. A. and Wafula, B. M. 1992. Modelling the fully-expanded area of maize leaves. Field crops Research, 29: 163-176.

- Carberry, P. S.; Muchow, R. C. and McCown, R. L. 1989. Testing the CERES-Maize simulation model in a semi-arid tropical environment. Field Crops Research, 20: 297-315.

- Carberry, P. S. and Abrecht, D. G., 1991. Tailoring crop models to the semi-arid tropics: In: RC Muchow and JA Bellamy (Eds) Climatic risk in crop production: Models and management in the semi-arid tropics and sub-tropics. Cab International, Wallingford . P. 157-182.

- Jones, C. A.and Kiniry, J. R. 1986. CERES-Maize: A simulation model of maize growth and development. Texas A & M University Press, College Station , texas, 194pp.

- Wilson, D. R.; Muchow, R. C. and Murgatroyd, C. J. 1995. Model analysis of temperature and solar radiation limitations to maize potential productivity in a cool climate. Field Crops Research, 43: 1-18.

**RC:** *Some references should be revised in text and in the list of References.*

AR: Thank you for the comment. The following references were fixed in the main manuscript. Akhavizadegan et al (2021); Archontoulis et al (2014, 2020); Balboa et al (2019); Crane-Droesch (2018); Das et al (2020); Dietze et al (2013); Dietzel et al (2016); Flathers and Gessler (2018); Guerif and Duke (2000); Hoffman et al (2020); Jeong et al (2016); Kang et al (2020); van Klompernburg et al (2020); Leng and Hall (2020); Li et al (2014); Martinez-Feria et al (2019); Pasley et al (2021); Puntel et al (2016); Shahhosseini et al (2021); Spijker et al (2021); Wallach et al (2021), Vergopolan et al (2021), Chakrabarti et al (2014)

---

## Author Response (AR1)

**Authors' Response to Reviews of**

**A comprehensive assessment of in situ and remote sensing soil moisture data assimilation in the APSIM model for improving agricultural forecasting across the U.S. Midwest**

Marissa Kivi, Noemi Vergopolan, and Hamze Dokoohaki
*Hydrology and Earth System Sciences (HESS),*
* * *
**RC:** *Reviewers' Comment*,     AR: Authors' Response,     ☐ Manuscript Text

**1. Reviewer #1**

**RC:** *Overall, the referee's comments are positive and no major concerns have been raised. Several questions have been asked, though, which I kindly ask you to answer carefully. Please also add clarifications to the manuscript where considered appropriate. In addition to the Referee comments, I kindly ask you to consider the following:*

**RC:** *L309: Please change references to equation symbols to math-mode and make superscripts consistent (do-a and do-f):*

 AR: Thank you for the comment. This was modified and now can be found in L324.

**RC:** *L339: "To set an upper bound": I would advise against calling the assimilation of SM sensor observations "ideal". First, there are representativeness errors present that are difficult to account for (which is the reason why the assimilation of RS observation has often been found to outperform sensor assimilation); and second, also SM sensors exhibit uncertainties that would need to be parameterised properly. While providing a good benchmark, I wouldn't consider it ideal or an upper bound.*

 AR: Thank you for the comment. The term "ideal" was replaced with "reasonable benchmark".

**RC:** *L343: The successive assimilation of these data sets will inevitably introduce error covariances between the model runs and the observations, especially when assimilating data sets from the same source (in fact, all of them are based on SMAP to some degree). I recommend discussing the limitations of this approach and the implication of neglecting error covariance.*

 AR: We agree with this assessment and therefore we added a paragraph discussing this point in the discussion section under section (4.3).

**RC:** *L361: Definitions are given for RMSE, R2, etc. but not for the spectral norm and the wieghted variance. Can you provide these?*

 AR: Thank you for the comment. These two definitions were added to the manuscript.

**RC:** *L406: The figure this discussion seems to be referring to does not show differences between site years, therefore discussions about that leaves the reader to trust the authors.*

Thank you for pointing this out. Although more figures could be added to support this statement, we decided to remove this claim in favor of keeping the same number of figures.

**RC:** *More generally: Many of the discussions are based on supplementary material. However, all figures and tables that are relevant/necessary to support your findings and understand your results should be part of the main manuscript body. I therefore kindly ask you to reconsider your choice as to which figures go into the main body, and which in the supplement.*

AR: Thank you for your input. We went through the manuscript and double-checked all references to all figures to ensure there were no problems. However, we are open to further editorial suggestions for moving the figures between the supplementary material and the main manuscript body.

**RC:** *L454: Referring to "Fig. 6a" but there is no annotation of (a) in the Figure.*

AR: Thank you for the comment. (a) was removed from the text.